# How Do Cities Flow in an Emergency? Tracing Human Mobility Patterns during a Natural Disaster with Big Data and Geospatial Data Science

**Su Yeon Han** [1,*] **, Ming-Hsiang Tsou** [2] **, Elijah Knaap** [1] **, Sergio Rey** [1] **and Guofeng Cao** [3]

[1]   Center for Geospatial Sciences, University of California, Riverside, CA 92521, USA;
      knaap@ucr.edu (E.K.); sergio.rey@ucr.edu (S.R.)
[2]   Center for Human Dynamics in the Mobile Age, San Diego State University, San Diego, CA 92182, USA;
      mtsou@mail.sdsu.edu
[3]   Department of Geosciences, Center for Geospatial Technology, Texas Tech University,
      Lubbock, TX 79409, USA; guofeng.cao@ttu.edu
*   Correspondence: suhan@ucr.edu

**Abstract:** Understanding human movements in the face of natural disasters is critical for disaster evacuation planning, management, and relief. Despite the clear need for such work, these studies are rare in the literature due to the lack of available data measuring spatiotemporal mobility patterns during actual disasters. This study explores the spatiotemporal patterns of evacuation travels by leveraging users' location information from millions of tweets posted in the hours prior and concurrent to Hurricane Matthew. Our analysis yields several practical insights, including the following: (1) We identified trajectories of Twitter users moving out of evacuation zones once the evacuation was ordered and then returning home after the hurricane passed. (2) Evacuation zone residents produced an unusually large number of tweets outside evacuation zones during the evacuation order period. (3) It took several days for the evacuees in both South Carolina and Georgia to leave their residential areas after the mandatory evacuation was ordered, but Georgia residents typically took more time to return home. (4) Evacuees are more likely to choose larger cities farther away as their destinations for safety instead of nearby small cities. (5) Human movements during the evacuation follow a log-normal distribution.

**Keywords:** human movement; Hurricane Matthew; tropical cyclones; evacuation planning; evacuation travel; geospatial data science; big data; Twitter

## 1. Introduction

Tropical cyclones including hurricanes and typhoons have caused tremendous economic losses, human suffering, and related vast consequences that last for several years [1–3]. Among all the natural disaster losses in the United States (U.S.), the percentage of losses from tropical cyclones has been increasing since 2000 and occupied 41% in 2015 [4]. Although there is increased risk in coastal areas associated with hurricanes, humans have continued to migrate to coastal areas in the U.S. According to the National Coastal Population Report [5], "regardless of how the coast is defined, it is substantially more crowded than the U.S. as a whole, and population density in coastal areas will continue to increase in the future." Because the population is concentrated along coastal areas, disastrous hurricanes in these areas have often required massive evacuation, which creates human migration. In this situation, it is critical to understand when, where, and how residents in evacuation zones move during hurricanes, because this knowledge can reduce traffic congestion during the evacuation, avoid casualties and

economic damage during the storm, and assist in emergency response planning and post-disaster relief [6–9].

The modern era of big data provides a variety of opportunities to study the spatiotemporal patterns of human movements that the traditional methods and datasets (e.g., census) cannot reveal; however, there have been very few attempts to study human movements during disaster evacuation. Recent studies have provided improved knowledge about human mobility by utilizing mobile phone data [10–13] and social media data [14–17]. These early studies were groundbreaking because they demonstrated the ability to measure population change and movements at unprecedentedly high spatial and temporal resolution. In more recent work, Wang and Tayler [18] used Twitter data to examine human mobility patterns within New York City during Hurricane Sandy in 2012, and Martin et al. [19] used Twitter data to estimate the percentage of evacuees during Hurricane Matthew in 2016. Despite the growing use of Twitter data in disaster research; however, there is a dearth of work that examines how these data can provide insight into the spatiotemporal mobility patterns of evacuation zone residents over the duration of a hurricane evacuation. Filling this gap is critical because evacuation zones are among the greatest impacted locations during an emergency, and better understanding, modeling, and planning of evacuations can have a dramatic effect.

To fill this gap, this research uses data from millions of geotagged tweets to analyze the spatiotemporal mobility patterns of Twitter users residing in southeast coastal evacuation zones during Hurricane Matthew, the strongest and deadliest storm in the Atlantic in 2016. Our work provides considerable evidence that geotagged tweets are a valuable resource for analyzing evacuees' travel patterns during Hurricane Matthew and other natural disasters. With such data, we are able to do the following: measure whether, where, and how long evacuees left disaster zones; characterize the location decisions of evacuees, including whether they remained near evacuation zones or traveled to other small or large cities in the region; and validate the utility of our data by demonstrating whether the evacuation rate can be estimated using Twitter data alone.

The remainder of the paper is organized as follows. Section 2 provides an overview of Hurricane Matthew and reviews prior research related to studying human mobility with Twitter data and the characteristics of spatial information included in Twitter data. Section 3 describes our procedures for data collection, locating tweets, and noise filtering and our methods for visualizing spatiotemporal mobility patterns using Twitter data. Section 4 presents the results of a model that estimates our observed movement patterns, and the paper concludes with a discussion of the key findings and limitations of the study.

## 2. Background

### 2.1. Hurricane Matthew

Hurricane Matthew was formed near the Windward Islands in late September 2016 [20] and reached the eastern Caribbean three days later. On October 1, the intensity of Matthew reached its peak at Category 5. Then, Matthew, as a Category 4, made landfall in Haiti on October 4. After that, Matthew hit the Bahamas on October 5–6 and the southeastern U.S. as it moved close to coastal areas of Florida, Georgia, South Carolina, and North Carolina [21]. In Florida, more than 1.5 million people were under evacuation orders [22]. Although they were not directly hit by Matthew, storm surges caused massive flooding in Florida [23]. As a result, more than 1 million Florida residents were without power [23]. Between October 6 and 7, Matthew was downgraded from Category 4 to Category 2 and struck the Georgia coast [24]. About 500,000 residents in Georgia were ordered to evacuate their homes [24]. Although many cities in Georgia, including Savannah, avoided a direct hit from Matthew, the storm surge caused flooding and power outages over a large area [23]. Matthew finally made an official landfall on the morning of October 8, 40 miles northeast of Charleston, South Carolina, as a Category 1 hurricane with 85 mph winds [20]. Charleston faced massive storm surges and widespread flooding [23]. Piers at Myrtle Beach were destroyed, and at least 800,000 lost power [23].

About 1 million people were told to evacuate along the coastal counties of South Carolina [25]. North Carolina also had extensive flood damage, and about 760,000 remained without power [23]. Thousands of people in North Carolina were urged to leave their homes, especially in low-lying areas [25]. According to *USA Today*, the economic damage of Hurricane Matthew is estimated to be between $4 and 6 billion in the U.S. [21]. Matthew killed a total of 43 people in the U.S.: 12 deaths in Florida, three deaths in Georgia, 26 deaths in North Carolina, and two deaths in Virginia [26].

In terms of evacuation planning, the evacuation order was issued for residents in coastal areas of Florida, Georgia, South Carolina, and North Carolina in advance of the storm. The evacuation order was given to 1.5 million residents in evacuation zones on the Atlantic coast of Florida on October 5 and 6. In Georgia, on October 5, the governor ordered a mandatory evacuation of six coastal counties due to the potential effects of Hurricane Matthew [27]. In South Carolina, the mandatory evacuation order was given by the governor on October 4 for residents in coastal counties, and they began their evacuation the next day [28].

The human movement caused by evacuation during Hurricane Matthew gives us an opportunity to study human movement patterns in emergency situations. On the other hand, the prevalence of geotagged social media data, such as Twitter data, provides us an opportunity to study human movements based on spatial footprints that are created from global positioning system (GPS)-enabled smartphones. Based on these two research opportunities, this study utilizes Twitter data to study evacuation-related travels during Hurricane Matthew.

### 2.2. The Use of Social Media Data to Study Human Mobility Patterns

Researchers have examined whether Twitter data can reveal patterns of actual human movements [15,16,29]. Hawelka et al. argue that international travel patterns that are identified based on users' locations in Twitter data can be representative of all international air traffic [15]. Jurdak et al. show that coordinates of Twitter users' locations can be used to examine different trajectories of some users and their travel patterns within and between cities [16]. Liu et al. revealed the correlation between the distributional patterns of Twitter users and the distribution of the population of the census [29]. Those previous studies have demonstrated that geotagged tweets can reveal patterns of human movements. However, studies rarely discuss the practical application of Twitter data as a proxy of human travels. Moreover, there have been few attempts to examine whether Twitter data can reveal the spatiotemporal patterns of evacuees during disastrous events. Unlike the previous studies, this study suggests the practical use of Twitter data to explore evacuees' travel patterns during natural disasters, particularly Hurricane Matthew.

In terms of studying human movement patterns during the disaster, Martín et al. utilized geo-tagged Twitter data and quantified evacuation participants and compliance by residents in South Carolina during Hurricane Matthew [19]. This study specifically examined the timing of departure after the evacuation order and re-entering after the hurricane and quantified the destinations during evacuation travel at the state level. Unlike the previous study, our study focuses on examining the spatiotemporal travel patterns of evacuees by utilizing Twitter messages. We chose evacuation areas in Georgia and South Carolina as study areas. Our study also differs from prior work in our methods for noise filtering Twitter messages, identifying evacuees among Twitter users, spatiotemporal resolution, and visualization. Furthermore, unlike prior work, we model the distribution of Twitter users' travel destinations during the period of evacuation, identify the direction of travel by utilizing flow maps, and examine the daily changing distributional patterns of Twitter users' movements during the evacuation.

### 2.3. Twitter Data Characteristics Related to Users' Locations

Twitter is a social networking service through which registered users broadcast short messages called tweets. Twitter users can broadcast tweets and follow other users' tweets by using mobile devices or desktops. By utilizing the Twitter application programming interfaces (APIs), researchers can download users' locations, which are recorded in three different fields in Twitter data: 'coordinates',

'bounding box', and 'location.' The entire fields of Twitter data are available on the Twitter website [30]. Some Twitter data contain coordinates of users' locations when they post their messages with GPS-enabled mobile devices. Some Twitter users' locations are also recorded in the form of 'bounding box' when they enter a 'Tag location' with their messages. When users select 'Tag location', Twitter suggests some of the places that they can select as their locations. For example, the locations include city names, such as San Diego, or points of interests, such as the names of stadiums, restaurants, malls, grocery stores, etc. Once the user selects one of the suggested places, coordinates for each of four corners of a bounding box (rectangle), which encloses the selected place, is included in Twitter data. The field called 'location' in Twitter data also provides some information about users' locations because it mostly uses the city name that the users entered in their profiles. Unlike the 'coordinates' and 'bounding box', 'location' in some cases does not show users' current locations, because the actual tweeting locations and the city names that users entered as their residences are not always same.

## 3. Research Design and Method

This study consists of the following five procedures (Figure 1). Each subsection describes the tasks in each step.

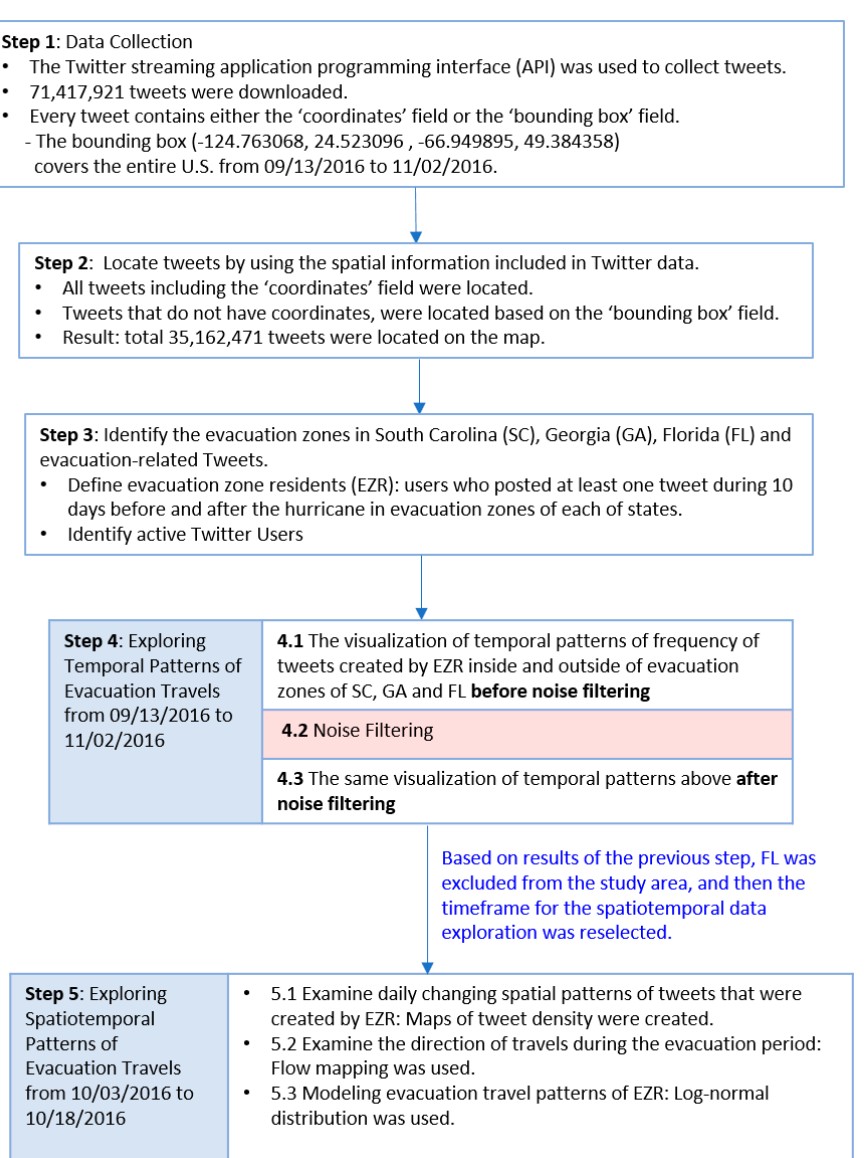

**Figure 1.** The workflow of this study, including methods that were used for the analysis.

### 3.1. Step 1: Data Collection

We used the Twitter streaming API and downloaded all tweets by using the bounding box (−124.763068, 24.523096, −66.949895, 49.384358) that covers the entire U.S. from September 13, 2016 to November 2, 2016. The hurricane directly affected the U.S. for only a few days. However, we examined the data over 1.5 months because we wanted to examine the spatial-temporal patterns of Twitter users before, during, and after the hurricane.

### 3.2. Step 2: Locating Tweets

This step examines the geographic information of the Twitter data collected in step 1, exploring, in particular, how accurately users' locations can be located for the next step of the spatiotemporal analysis. In the data collected by the Twitter streaming API, every tweet has the fields of metadata, 'coordinates', or 'bounding box.' The 'coordinates' field contains the longitude and latitude geographic coordinates of users' locations in decimal degrees. However, not every tweet has coordinates. Only 11.34% of all the tweets studied here have users' coordinates (see the first row in Table 1). Tweets without coordinates have a 'bounding box' field. The 'bounding box' has longitude and latitude coordinates that enclose the places where users are located. The size of the bounding box varies: the diagonal length of the bounding boxes ranges from a few kilometers to a few thousand kilometers. Table 1 shows the number of tweets per each different size of bounding box. The second row of Table 1 represents the tweets containing the coordinates of users (the exact locations), so the diagonal length of the bounding box is represented as 0. For the tweets containing bounding boxes without the exact users' locations, the number of tweets per each interval was counted from the third row of Table 1. For example, there are 1,951,799 tweets whose diagonal length of the bounding boxes was greater than 0 and less than 5 km and so on. As Table 1 indicates, Twitter data for which users' exact locations are known only cover about 11% of the entire dataset. Twitter data having less than 20 km diagonal length of the bounding boxes cover about 50% of the entire data set. Similarly, Twitter data with less than 30 km diagonal length of the bounding boxes cover about 63% of the entire data. Very rarely, the diagonal length is huge, such as a few thousand kilometers (see the last row of Table 1).

**Table 1.** The number of tweets depending on the different size of the bounding boxes. Column (**A**) represents the diagonal length of the bounding boxes. Column (**B**) represents the number of tweets per each range of different size of bounding box in column (**A**). Column (**C**) represents the percentage of tweets in each interval over the total number of tweets. Column (**D**) is the percentage of the cumulative tweets in each interval over the total number of tweets.

| (A) The Diagonal Length of Bounding Boxes (km) | (B) The Number of Tweets | (C) Percentage | (D) Cumulative Percentage |
|---|---|---|---|
| 0 | 8,098,058 | 11.34 | 11.34 |
| 0–5 | 1,951,799 | 2.73 | 14.07 |
| 5–10 | 8,297,873 | 11.62 | 25.69 |
| 10–20 | 16,814,741 | 23.54 | 49.23 |
| 20–30 | 10,315,984 | 14.44 | 63.68 |
| 30–50 | 7,543,440 | 10.56 | 74.24 |
| 50–100 | 7,014,155 | 9.82 | 84.06 |
| 100–150 | 1,523,280 | 2.13 | 86.2 |
| 150–200 | 175,364 | 0.25 | 86.44 |
| 200–300 | 262,287 | 0.37 | 86.81 |
| 300–500 | 423,922 | 0.59 | 87.4 |
| 500–1000 | 6,880,665 | 9.63 | 97.04 |
| 1000–3000 | 2,116,353 | 2.96 | 100 |

To locate each user, we first used tweets containing coordinates, which are the 8,098,058 tweets in the second row of Table 1. We also considered tweets whose diagonal length of the bounding box is less

than 20 km. We assumed that each user was located at the centroid of each bounding box. The 20-km threshold was determined through trial and error. When we considered tweets only containing the exact coordinates (11% of all tweets), the number of tweets was too small to capture enough movements between cities. On the other hand, it is not meaningful to locate tweets with huge bounding boxes because of the great uncertainty about the location of users within these bounds. For this reason, the 20-km length of the diagonal box was chosen as the threshold, i.e., we located the tweets whose diagonal length of the bounding box is less than 20 km at the centroid of the respective bounding box and discarded tweets whose diagonal length of the bounding box is larger than 20 km. In Figure 2, a 10-km buffer from the evacuation zone was drawn to show the relative size of the 10-km buffer compared with the size of the evacuation zones on the map.

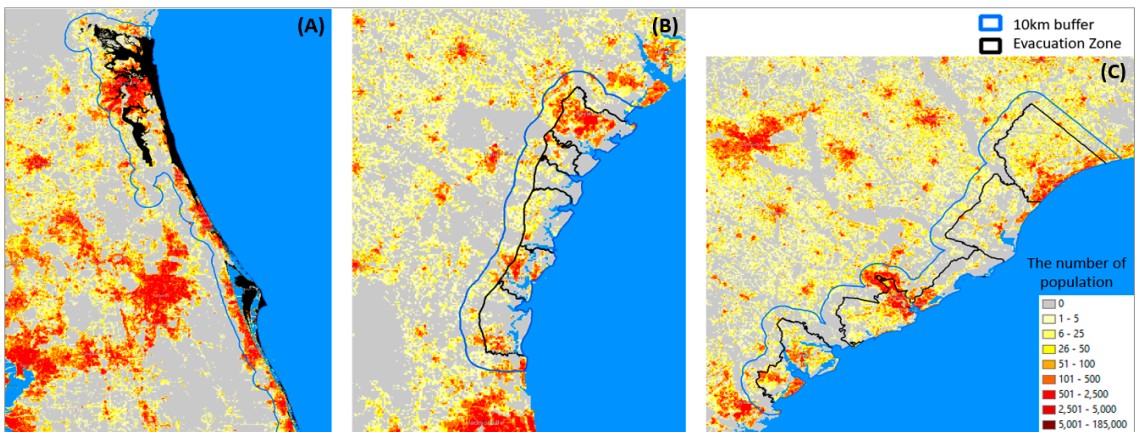

**Figure 2.** Evacuation zones during Hurricane Matthew in Florida (**A**), Georgia (**B**), and South Carolina (**C**). Evacuation zones were filled with black in Florida (**A**), because the zones are too narrow to be represented by boundaries. Landscan 2015 was used as a base layer to show the population distribution.

### 3.3. Step 3: Identifying Evacuation Zones and Tweets Related to Evacuation Travel

In this step, we first decided which regions we were going to examine to capture evacuation-related travels during Hurricane Matthew. We initially aimed to examine evacuation travels from the evacuation zones in four U.S. states, North Carolina (NC) South Carolina (SC), Georgia (GA), and Florida (FL). However, while exploring temporal patterns of Twitter data (described in the next step), we selected SC and GA, and in these two states, we examined the spatiotemporal patterns of evacuation travels (The reason for this decision is described in Section 4.2). Figure 2 shows evacuation zones during Hurricane Matthew in SC, GA, and FL. Evacuation zones in these three states were identified from the Weather Channel [13]. There is relatively less information available about evacuation zones in NC compared with those in the other states. According to CBS News [31], the evacuation zone during Hurricane Matthew in NC was along the Neuse River in Lenoir and Kinston County.

We defined evacuation zone residents (EZR) as users who posted at least one tweet in the 10 days before the hurricane (between September 24 and October 3, 2016) and at least one tweet in the 10 days after the hurricane (between October 18 and October 27, 2016) in the evacuation zones. If a user tweeted at least once before and after the hurricane in the evacuation zones, we assumed that this user is a resident of the evacuation zones. A user's residency can be short term, such as only a few weeks, in the evacuation zones, or it can be a long-term residency. Evacuation travel-related tweets are the tweets created by EZR. By tracking locations of tweets created by EZR, it was possible to see trajectories of EZR's movements during evacuations.

*3.4. Step 4: Exploring Temporal Patterns of Evacuation Travel*

3.4.1. Visualization of Temporal Patterns of the Frequency of Tweets Created Inside and Outside of Evacuation Zones by Evacuation Zone Residents (EZR)

This step focuses on comparing the distribution of daily tweets from inside and outside of the evacuation zones. First, we visualized the daily frequency of tweets created by EZR in two separate regions, inside and outside of the evacuation zones in each state: SC, GA, and FL. We hypothesized that there would be a reversed pattern of the distribution of tweets from inside and outside of the evacuation zones, i.e., EZR would post most of their tweets within the evacuation zones before and after the evacuation period while posting most of their tweets outside of the evacuation zones during the evacuation period. While examining the distribution of tweets, we found that Twitter data containing noises could not reveal the reversed pattern. Therefore, in the next step, after the noises were filtered out, the distribution of tweets was examined once again. The detailed description of the method for noise filtering is described below.

3.4.2. Separating Noise from Tweets Created by Human Users

This step focuses on filtering out the noise in Twitter data. To filter out the noise, we used the source list identification method (SLIM), developed by the Center of Human Dynamics in the Mobile Age [32]. SLIM aims to separate non-meaningful tweets (i.e., noise) such as job advertisement, traffic, and weather broadcasting from meaningful tweets that contain human expressions, such as sharing activities and thoughts. Non-meaningful tweets are mostly posted by non-human users (i.e., robots), and meaningful tweets are tweets posted by human users. SLIM suggests that non-meaningful tweets can be decided based on the source of Twitter data, because the sources that create non-meaningful tweets tend to create non-meaningful tweets constantly. Sources in Twitter data mean platforms through which tweets are created. For example, in Table 2, the first three tweets were created through a source called "Ebb Tide Bot." Through the same source, thousands of tweets were created in the dataset we collected. All tweets created through this platform are similar to the first three tweets in Table 2, which broadcasts tidal information.

The number of sources is relatively small compared with the number of tweets. For example, in our case study, millions of Twitter messages were created from around 50 different sources. Therefore, we manually examined the content of a few tweets from each different source to identify sources that generate non-meaningful tweets. Specifically, in this study, 1,066,278 tweets were created through 49 different sources. Therefore, we manually examined only two or three tweets for each of these 49 different sources to differentiate sources that create meaningful tweets from non-meaningful tweets. Through this manual process, we classified the 49 different sources into two categories: (1) sources that generate meaningful tweets and (2) sources that generate non-meaningful tweets. Then, we made a computer program that (1) automatically checks if the source of each of tweet is one that constantly creates non-meaningful tweets or not and (2) divides all tweets into two files: one file that contains tweets created by sources generating meaningful tweets and another file that contains tweets created by sources generating non-meaningful tweets. After this step, only tweets in the first file were used for the visualization and analysis in the next steps. Table 2 shows examples of Twitter messages created through a platform that creates non-meaningful tweets. A list of all sources creating non-meaningful tweets is available in Table S1.

**Table 2.** Examples of tweets created by non-human users. 'Ebb Tide Bot' broadcasts tidal information. 'World Cities' broadcasts weather conditions. 'TweetMyJOBS' and 'Safe Tweet by TweetMyJOBS' advertise jobs. 'circlepix' advertises housing. 'Simply Best Coupons' advertises restaurants, tours, and events by offering coupons.

| Sources (Platforms through which Tweets Are Created) | Tweets (Text) |
|---|---|
| Ebb Tide Bot | @ebbtideapp Tide in Summit Bridge, Delaware 09/12/2016 Low 1:38am 0.5 High 7:41am 3.4 Low 1:38pm 0.4 High 7:59pm 3.9 |
| | @ebbtideapp Tide in Hobcaw Point, South Carolina 09/12/2016 Low 10:51pm 1.1 High 4:44am 5.2 Low 10:52am 0.7 High 5:33pm 6.0 |
| | @ebbtideapp Tide in Quonset Point, Rhode Island 09/12/2016 Low 10:23pm 0.8 High 4:24am 3.3 Low 10:13am 0.7 High 4:53pm 3.7 |
| TweetMyJOBS | Want to work at HMSHost? We're #hiring in #Savannah, GA! Click for details: https://t.co/1voU52Qtdq #Job #Hospitality #Jobs #CareerArc |
| | This #job might be a great fit for you: Carhop/Skating Carhop (Server) - https://t.co/DD72Z53XCn #SONIC #Hospitality #Savannah, GA #Hiring |
| | Want to work in #Savannah, GA? View our latest opening: https://t.co/NO4klr9fzr #Job #Hospitality #Jobs #Hiring #CareerArc |
| World Cities | current weather in Brunswick: clear sky, 87F 74% humidity, wind 10mph, pressure 1017mb |
| | temperature down 87F -&gt; 81F humidity up 74% -&gt; 83% wind 10mph -&gt; 5mph |
| | clear sky -&gt; scattered clouds temperature down 81F -&gt; 77F humidity up 83% -&gt; 100% wind 5mph -&gt; 4mph |
| circlepix | Check out my #listing in #WhiteOak #GA #realestate #realtor https://t.co/4kWMMnUu94 https://t.co/pa76kCSOZy |
| | Just Listed in #Cumming #GA. 3320 Sweetwater Dr! Please retweet! https://t.co/u7Eaw04zcQ https://t.co/uaUPxDWjU4 |
| | I would love to show you my #listing at 5956 Harrietts Bluff #Woodbine #GA #realestate https://t.co/OmYJi5Y9L3 https://t.co/bO8DYpbHxX |
| SafeTweet by TeetMyJOBS | This #job might be a great fit for you: STORE MANAGER CANDIDATE in Victoria TX - https://t.co/nvCb5mR33N #Retail https://t.co/sb9j2vBTpG |
| | Can you recommend anyone for this #job in #MOUNTAINTOP, PA? https://t.co/m9Yj6Ufmc7 #Diversity #Retail #Hiring https://t.co/EkrkakKldW |
| | We're #hiring! Click to apply: STORE MANAGER CANDIDATE in HUDSON FALLS NY - https://t.co/3lINmZn6RJ #Diversity https://t.co/dVBfZbgl7t |
| Simply Best Coupons | Up to 38% Off Dolphin Cruise https://t.co/rCdXjPZh9N |
| | 42% Off Trolley Tours https://t.co/rCdXjPZh9N |
| | Up to 40% Off at FULL Lunch and Late Night https://t.co/W54t4RmCzt |

*3.5. Step 5: Exploring Spatiotemporal Patterns of Evacuation Travel*

3.5.1. Visualization of Daily Changing Spatial Patterns of Tweets created by Evacuation Zone Residents (EZR)

In this step, the distributions of daily changing tweets created by EZR in South Carolina and Georgia were visualized. Then, kernel density estimation (KDE) was used to show the distributional patterns of daily tweets. At this stage, the top destination cities of evacuation travels of EZR in each state were identified. Evacuation rates of Twitter users in each state were also computed.

3.5.2. Visualization of the Direction of Travel during the Evacuation Period

The flow mapping method was used to show the direction of travel. We hypothesized that in the days just before and during the hurricane, most of the travels of EZR would be from evacuation zones to outside evacuation zones, and in the days just after the hurricane, most of the travels of EZR would be from outside evacuation zones to inside evacuation zones. Therefore, we created flow maps showing flow lines between origins and destinations for two separate periods, the pre-hurricane period (from the start of the evacuation until the evacuation order was lifted) and the post-hurricane period

(from when then evacuation order was lifted until a few days after the evacuation order was lifted). In addition, to reveal the direction of travel, we classified the travels of EZR into three categories: inflow (i.e., travels from outside evacuation zones to inside evacuation zones), outflow (i.e., travels from inside evacuation zones to outside evacuation zones), and the rest of flows, which are travels that do not cross the boundary of the evacuation zones. On flow maps, these three different types of travels were represented in three different colors (Figure 3). When a Twitter user moved across the boundary of the evacuation zone from inside to outside the evacuation zone, this movement is represented by a red line (outflow). When a user moved across the boundary of the evacuation zone from outside to inside the evacuation zone, this movement is represented by a green line (inflow). The rest of the movements that did not cross the boundary of the evacuation zone are represented by grey lines.

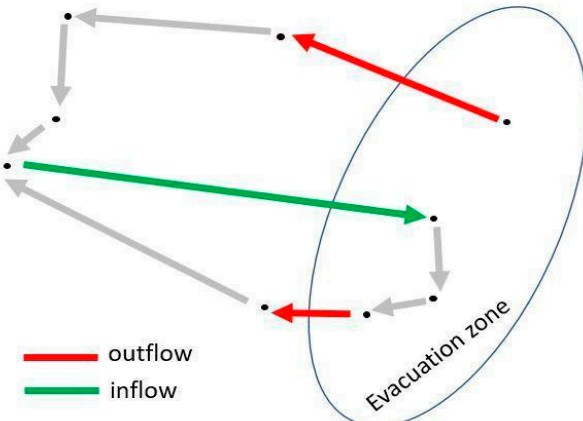

**Figure 3.** An example trajectory of a Twitter user. Black dots are locations where the user posted a tweet. Arrows were not added in the resulting maps.

Based on the tweeting locations of users, it is challenging to detect inflows and outflows of their evacuation travel because people do not tweet consistently. After step 3 in Section 3.3, we identified tweets of users who tweeted at least once during the 10 days before and after the Hurricane in the evacuation zones. It would be easier to detect inflows and outflows of their evacuation travels if people tweeted at least once right before and after evacuation travel and at their final destinations. However, in reality, some people might tweet one or two days before they make a trip; they might not tweet at their final destinations, but they may tweet again a few days after they come back from their evacuation travels. In this case, we cannot identify the inflows and outflows of their evacuation travels. However, if they tweeted at least once outside of the evacuation zones during the evacuation period, it is possible to identify inflow and outflow. For this reason, we could not identify all evacuation-related travels of Twitter users. However, the goal was to identify as many of the inflows and outflows as we could. In this study, we set October 3–8 as the pre-hurricane period and October 8–13 as the post-hurricane period. Pre-hurricane was defined to capture outflow, and post-hurricane was defined to capture inflow.

### 3.5.3. Modeling Evacuation Travel Patterns of EZR

This step examines which model best fits the data representing human movements during the evacuation. We used the power law and log-normal distribution to model the distribution of displacement between origins and destinations during the evacuation travel. The power law distribution has the following form:

$$y = ax^b \tag{1}$$

where $a$ is a constant and $b$ is the power law exponent governing the shape of the distance decay curve. Moreover, the log-normal distribution is given by:

$$y = y_0 + \frac{A}{\sqrt{2\pi}wx}e^{\frac{-[\ln\frac{x}{xc}]^2}{2w^2}} \tag{2}$$

where $y_0$ = offset, $xc$ = center, $w$ = log standard deviation, and $A$ = area. In both equations above, y is the frequency of tweets, and x is the distance between a pair of tweets posted by the same person, i.e., origins and destinations. To measure the distance between them, locations of origins and destinations of evacuation travels of EZR were estimated. Because Twitter users do not tweet every time they visit places, Twitter data do not show every trajectory of users' movements. For the same reason, it is not easy to differentiate final destinations from intermedia stopping places. Due to the limitation of data, we assume that origins of EZR are the last locations where they tweeted in the evacuation zone before they left the evacuation zone, and destinations are every location where every EZR tweeted. The goodness of fit of the two models (Equations (1) and (2)) was measured by R-squared ranging from 0 to 1, 0 and 1.

## 4. Findings and Interpretation

### 4.1. The Result of Data Preprocessing and Noise Filtering

Table 3 shows the number of tweets created by EZR in South Carolina, Georgia, and Florida and the number of Twitter users who posted those tweets. Row e of Table 3 shows the number of tweets created by EZR of each of three states. In this case, EZR includes human users and non-human users. Therefore, the tweets created by human users (row a) were separated from the noise (row c) that was posted by non-human users to study human behaviors.

**Table 3.** The number of tweets created by evacuation zone residents and the number of users who created those tweets. Row (b) represents the number of users who created the number of tweets in row (a). Row (d) represents the number of user accounts that created the number of tweets in row (c). Row (e) represents the number of total tweets containing noise, and row (f) represents the number of users who created the number of tweets in a row (e).

|  | South Carolina (SC) | Georgia (GA) | Florida (FL) |
|---|---|---|---|
| (a) Tweets created by humans | 32,735 | 33,019 | 63,642 |
| (b) human users | 923 | 504 | 1179 |
| (c) noise | 346,343 | 247,004 | 343,535 |
| (d) users creating noise | 131 | 79 | 125 |
| (e) total tweets | 379,078 | 280,023 | 407,177 |
| (f) total users | 1054 | 583 | 1304 |

### 4.2. Effects of Noise Removal on Temporal Tweeting Patterns Inside and Outside of Evacuation Zones

Noise removal in tweets using the SLIM method [32] reveals unique temporal patterns in terms of tweeting frequency that were not able to be revealed before the noise removal in the case of South Carolina and Georgia. Figures 4A and 5A were created with the tweets containing noises. On the other hand, Figures 4B and 5B were created with the tweets after noise removal. With each different dataset, both Figure 4; Figure 5 describe the temporal change of tweeting frequency created by EZR inside and outside of the evacuation zones of SC and GA, respectively, from September 13, 2016 to November 2, 2016. The charts created with the tweets before the noise removal (Figures 4A and 5A) show that EZR of each state tweeted many more tweets outside of their residential areas (i.e., evacuation zones) than inside of the evacuation zones regardless of dates. However, Figure 4B shows that between October 6 to 10, which is during the evacuation period, EZR created fewer tweets inside evacuation zones and more tweets outside of the evacuation zones than during normal days—i.e., the days have nothing to do the hurricane evacuation. Similarly, Figure 5B shows that between October 6 and 16, EZR created fewer tweets inside evacuation zones and more tweets outside of the evacuation zones than the rest

of the days. The patterns in Figures 4B and 5B indicate that many Twitter users stayed outside of their residential areas (evacuation zones) for their safety from the hurricane, so they were not able to post tweets in the evacuation zones during the evacuation. In particular, during the period between October 7 and 9, the biggest number of tweets were created outside of the evacuation zones by EZR in both Figures 4B and 5B. The reason for this is that Hurricane Matthew made a landfall officially on October 8 in South Carolina.

On the other hand, the noise removal in tweets could not reveal the evacuation movements of EZR in Florida (Figure S1 in Supplementary Materials). Evacuation zones in FL were much narrower along the coastal areas than evacuation zones in South Carolina and Georgia (Figure 2). In this case, locating tweets with the 20-km diagonal length of bounding boxes was not accurate enough to capture movements between inside and outside the evacuation zones. (see Table 1 for issues with locating tweets). Therefore, we tried to create the same charts with tweets including only coordinates (Figure S2 in Supplementary Materials). However, in this case, the number of tweets was not enough to capture movements between inside and outside the evacuation zones. The same issue arises in terms of examining evacuation travels of EZR in North Carolina (NC). Evacuation zones in NC during Hurricane Matthew were narrow areas along the Neuse River in Lenoir and Kinston County. We found that Twitter data could not reveal movement patterns during hurricane evacuation in regions where the evacuation zones were narrow. On the other hand, in the case of SC and GA, their evacuation zones were county level or half county level. In this case, Twitter data could be used to reveal movement patterns during the hurricane evacuation. For this reason, NC and FL were excluded from the study area.

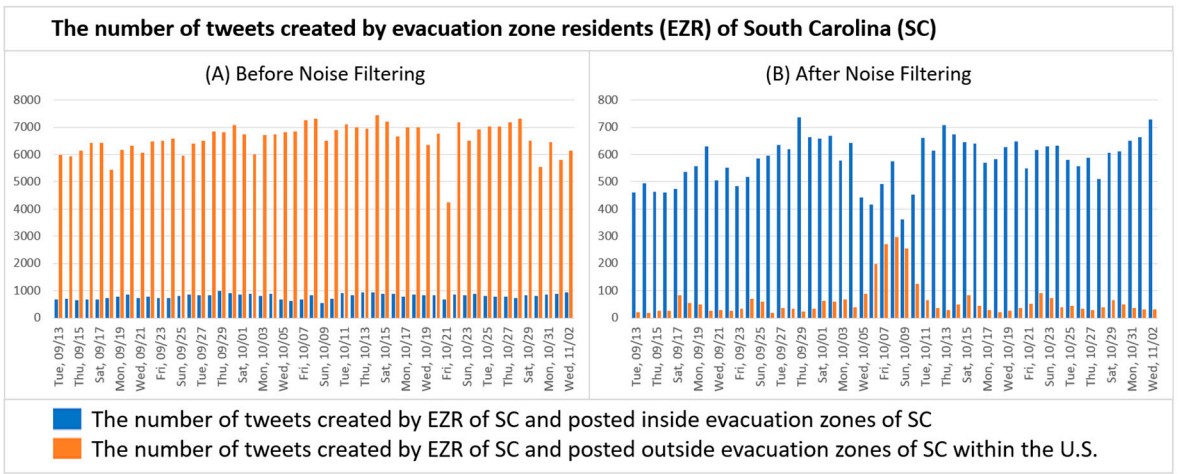

**Figure 4.** A comparison of tweeting frequency created by evacuation zone residents (EZR) of South Carolina (SC) before and after noise filtering inside and outside of the evacuation zones of SC in the U.S. Both charts represent the frequency of tweets created by users who created at least one tweet during the 10 days before the hurricane (September 24 to October 3) and at least one tweet during the 10 days after the hurricane (October 18 to October 27) in the evacuation zones of SC. (**A**) The frequency of tweets created by EZR of SC and posted inside and outside the evacuation zones of South Carolina before noise filtering. (**B**) The frequency of tweets created by EZR of SC and posted inside and outside the evacuation zones of SC after noise filtering. Tweets were located and used only if they contain coordinates or have a diagonal length of the bounding boxes of less than 20 km.

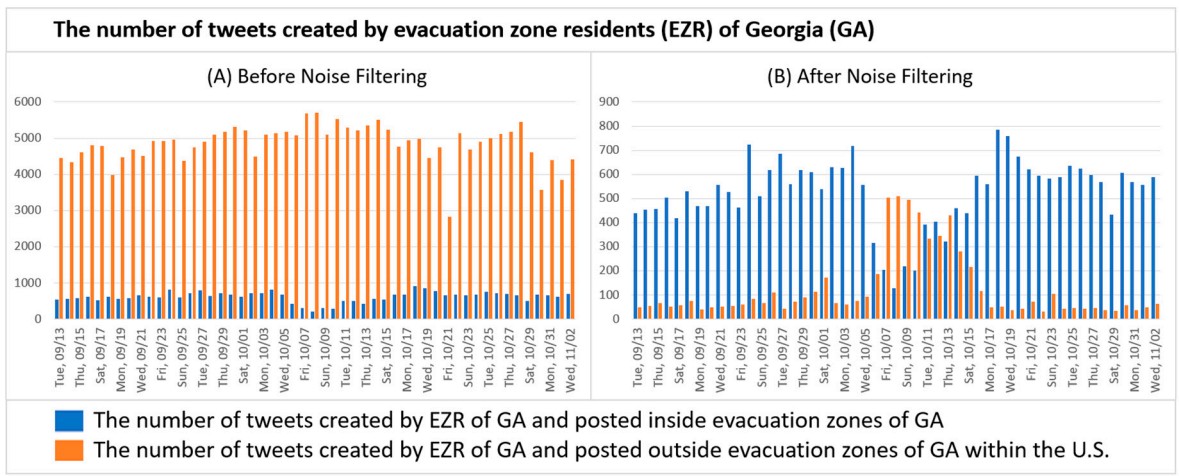

**Figure 5.** A comparison of tweeting frequency created by evacuation zone residents of Georgia (GA) before and after noise filtering inside and outside of the evacuation zones of GA in the U.S. Both charts represent the frequency of tweets created by users who created at least one tweet during the 10 days before the hurricane (between September 24 and October 3) and at least one tweet during the 10 days after the hurricane (between October 18 and October 27) in the evacuation zones. (**A**) The frequency of tweets created by EZR of GA and posted inside and outside evacuation zones of Georgia before noise filtering. (**B**) The frequency of tweets created by EZR of GA and posted inside and outside the evacuation zones of GA after noise filtering. Tweets were located and used only if they contain coordinates or have a diagonal length of the bounding boxes of less than 20 km.

### 4.3. Spatiotemporal Patterns of Evacuation Travels of Twitter Users

Twitter data reveal the spatiotemporal movement patterns of evacuation zone residents in South Carolina and Georgia. Maps in Figures 6 and 7 show the distribution of tweets created by EZR in SC and GA, respectively. One map per day was created by using kernel density estimation (KDE).

In terms of the spatiotemporal movement patterns of EZR in SC, tweeting densities in Figure 6 reveal that EZR started leaving the evacuation zones on October 5, 2016, and most of them came back to their town on October 12, 2016. As shown in Figure 6A,B, Twitter users mostly tweeted their tweets within the evacuation zones on October 3 and 4. Because these two days were before the storm onset, most EZR of SC tweeted in and around their residential areas, which were in the evacuation zones. On October 4, South Carolina's governor ordered a mandatory evacuation of coastal counties in SC. The next day, in response to the mandatory evacuation order, Twitter users started posting tweets outside of the evacuation zones, which means they started moving from the evacuation zones to outside of the evacuation zones (Figure 6C). From October 6 to 8 (Figure 6D–F), more and more tweets were created in places outside of the evacuation zones. Thus, as implied by Figure 6D–F, EZR gradually moved out of the evacuation zones several days in advance of the storm. On October 8, Hurricane Matthew made landfall in South Carolina. The next day, October 9, which was a Sunday morning, the evacuation order was lifted. Then, tweets created outside of the evacuation zones decreased from October 9 to 12 (Figure 6G–J). The decreasing number of tweets posted outside of the evacuation zones indicate that EZR gradually came back to the evacuation zones (their residential areas) after the evacuation order was lifted. In particular, on October 12 (Figure 6J), Twitter users posted most of their tweets in their residential areas, which means that most of them had returned from their evacuation travels. We also found that Figure 6A,B,J,K,O,P which are maps showing tweeting patterns during the normal weekdays, shows similar tweeting activities in terms of distributional patterns. The increased tweeting frequency outside of their residential areas (the evacuation zones) during the weekends from October 14 to October 16 (Figure 6L–N) can be interpreted as weekend trips outside of their towns during days off work.

The similar spatiotemporal movement patterns of EZR of Georgia are also shown in Figure 7. On October 3 and 4 (Figure 7A,B), Twitter users mostly tweeted their tweets within the evacuation zones, which were their residential areas. Thus, as shown in Figure 7A,B, EZR had not yet started evacuation travels. On October 5, Georgia residents east of Interstate 95 in six coastal counties were ordered to evacuate. In response to the evacuation order, as shown in Figure 7C, Twitter users started moving from their residential areas (the evacuation zones) to outside the evacuation zones on October 5. Between October 6 and 7, Hurricane Matthew struck the coast of Georgia. After the hurricane's onset, as shown in Figure 7D–L, more and more tweets were created in places outside of the evacuation zones from October 6 to 14. Unlike the case of South Carolina, in Georgia, Twitter users posted many of their tweets outside of the evacuation zones until October 14 (Figure 7L), even though the evacuation order was lifted on October 9, a Sunday night (Figure 7G). After October 15 (Figure 7M), the tweets created outside of the evacuation zone started decreasing. Then, as shown in the map of October 17 (Figure 7O), most of the tweets were created in the evacuation zones again, which means that most EZR finally came back to their towns in the evacuation zones. October 17 was the first day the tweeting patterns of EZR began to return to normal regardless of evacuation-related travel. Generally, Figure 7A,B,O,P shows the distributional patterns created by EZR for normal weekdays, and the rest of the maps show tweeting patterns related to the evacuation travel.

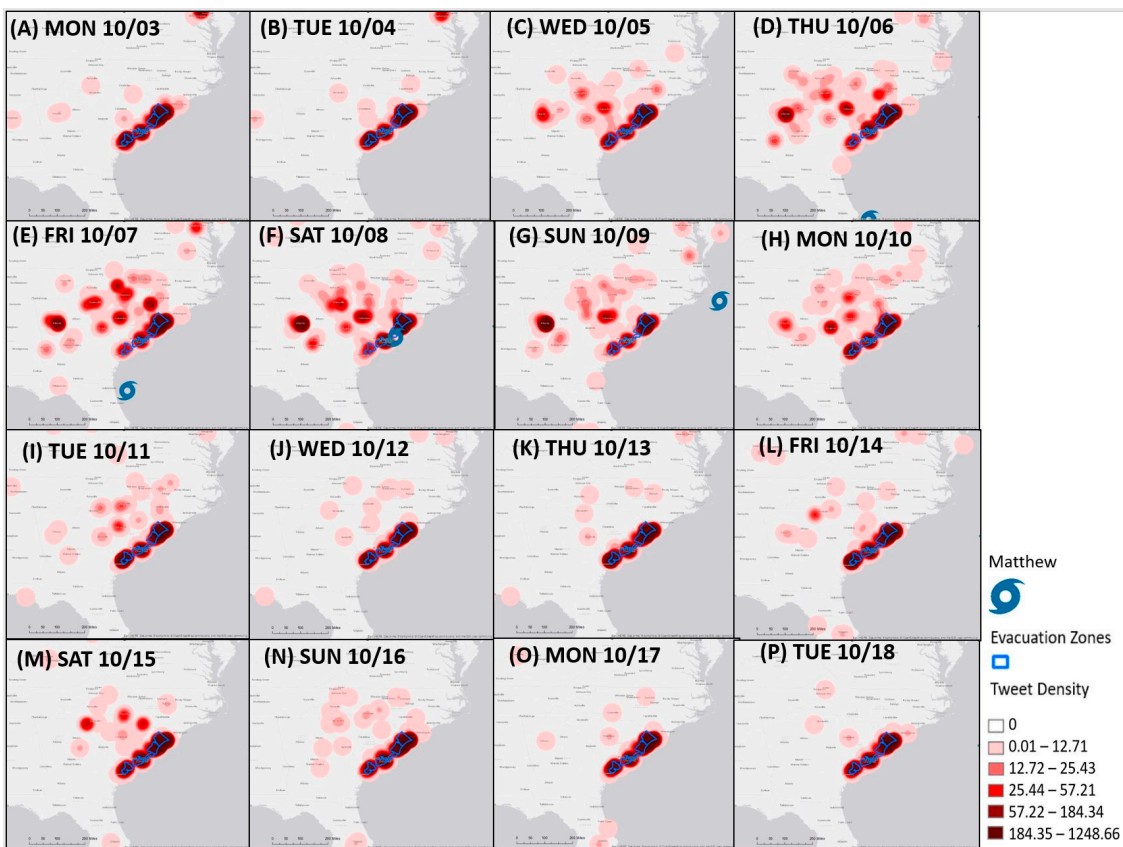

**Figure 6.** Spatiotemporal patterns of tweet densities created by evacuation zone residents in South Carolina. The maps represent the density of tweets that were posted by EZR. The mandatory evacuation zones are coastal counties, including Charleston, Beaufort, Horry, and Georgetown counties. Evacuation zones represented in blue borderline can be seen in the animated version [33].

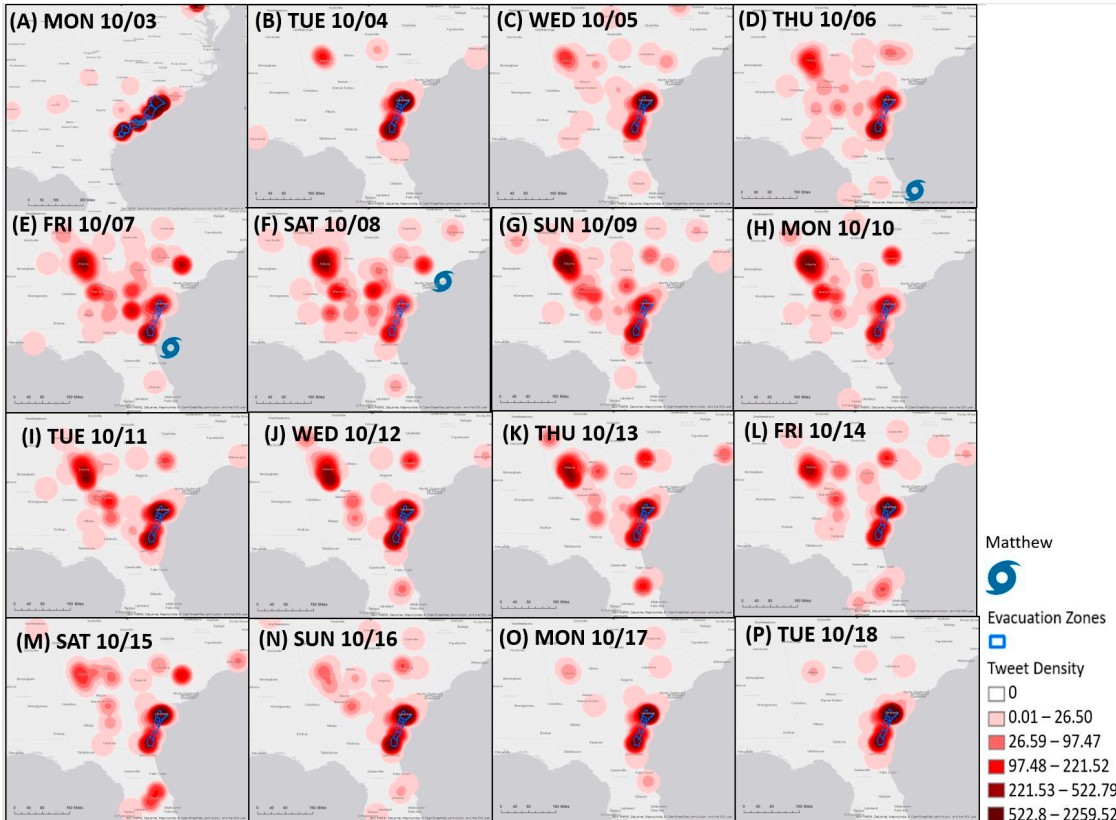

**Figure 7.** Spatiotemporal patterns of tweet densities created by evacuation zone residents in Georgia. The maps represent the density of tweets that were posted by EZR. The mandatory evacuation zones are six coastal counties east of Interstate 95 (I-95): Bryan, Chatham, Liberty, McIntosh, Glynn, and Camden. Evacuation zones represented in blue borderline can be seen in the animated version [34].

We found that a small portion of Twitter users created a large number of tweets, as shown in both Figures 6 and 7. As can be observed in Figure 6, 10,762 tweets were created by 815 users from October 3 to October 18. During the same period, as shown in Figure 7, 11,104 tweets were created by 456 users. We examined how many tweets were created per each individual user and found that in both cases, about 20% of tweets were created by 1% users. Some individual users created over 500 tweets from October 3 to October 18, and some users created only one tweet per person during the same period. In other words, the number of tweets created by each unique user shows a skewed distribution (Figure S3 in Supplementary Materials).

By utilizing the tweets created by evacuation zone residents during the evacuation period from October 6, 2016 to October 10, 2016 (Figure 6D–H and Figure 7D–H) the evacuation rates of Twitter users were computed. The motivation of the evacuation rate estimation is based on the research of Martin et al. [19], in which they gauged evacuation compliance by using Twitter data. Because we used a different approach in terms of filtering noise and identifying evacuation zone residents (see Sections 3.3 and 3.4.2 for details), we wanted to compare their estimation and our estimation for the evacuation rate. Our estimation shows that 37% of EZR in South Carolina and 62% of EZR in Georgia tweeted at least once outside of the evacuation zone during the evacuation period. In other words, 37% in SC and 62% in GA is our estimation of the evacuation rate. In the case of the evacuation rate in Georgia during Hurricane Matthew, no other estimation, such as a survey result, is available in order to validate the results of our estimation. However, in terms of the evacuation rate in SC, our estimation of 37% is similar to Cutter et al.'s estimation [35], which is 35% for a hypothetical evacuation, and is different from Martin et al.'s estimation, which is 54%.

### 4.4. Travel Directions of Evacuation Zone Residents among Twitter Users during the Hurricane Evacuation

Twitter data reveal the travel directions of evacuation zone residents in both the pre-hurricane and post-hurricane periods (see Section 3.5.2 for details). In the case of South Carolina, most Twitter users' travel directions were from evacuation zones to outside the evacuation zones during the pre-hurricane periods (October 3–8, 2016). Table 4 shows that 84.2% of travel crossing the boundary of the evacuation zones involved leaving evacuation zones from October 3 to 8, 2016 (pre-hurricane period). Figure 8 visually shows that most Twitter users left the evacuation zones after mandatory evacuation was ordered in the coastal counties (the pink-filled regions) on October 4, 2016 until the evacuation order was lifted on Sunday morning, October 9, 2016. On the other hand, Table 4 shows that 83.7% of travel crossing the boundary evacuation zones involved returning to residential areas from October 8 to 13, 2016 (post-hurricane period). Figure 9 visually shows that many evacuation zone residents returned to the evacuation zones (their residential areas) after the evacuation order was lifted on Sunday morning, October 9, 2016.

**Table 4.** The number and percentage of inflow and outflow in terms of travel of evacuation zone residents in South Carolina. See Figure 3 for the definition of inflow, outflow, and other movements.

|  | 10/03/2016–10/08/2016 | 10/08/2016–10/13/2016 | 10/08/2016–10/14/2016 |
| --- | --- | --- | --- |
| outflow (leaving) | 96 | 15 | 26 |
| inflow (returning) | 18 | 77 | 92 |
| % outflow | 84.21 | 16.30 | 22.03 |
| % inflow | 15.79 | 83.70 | 77.97 |

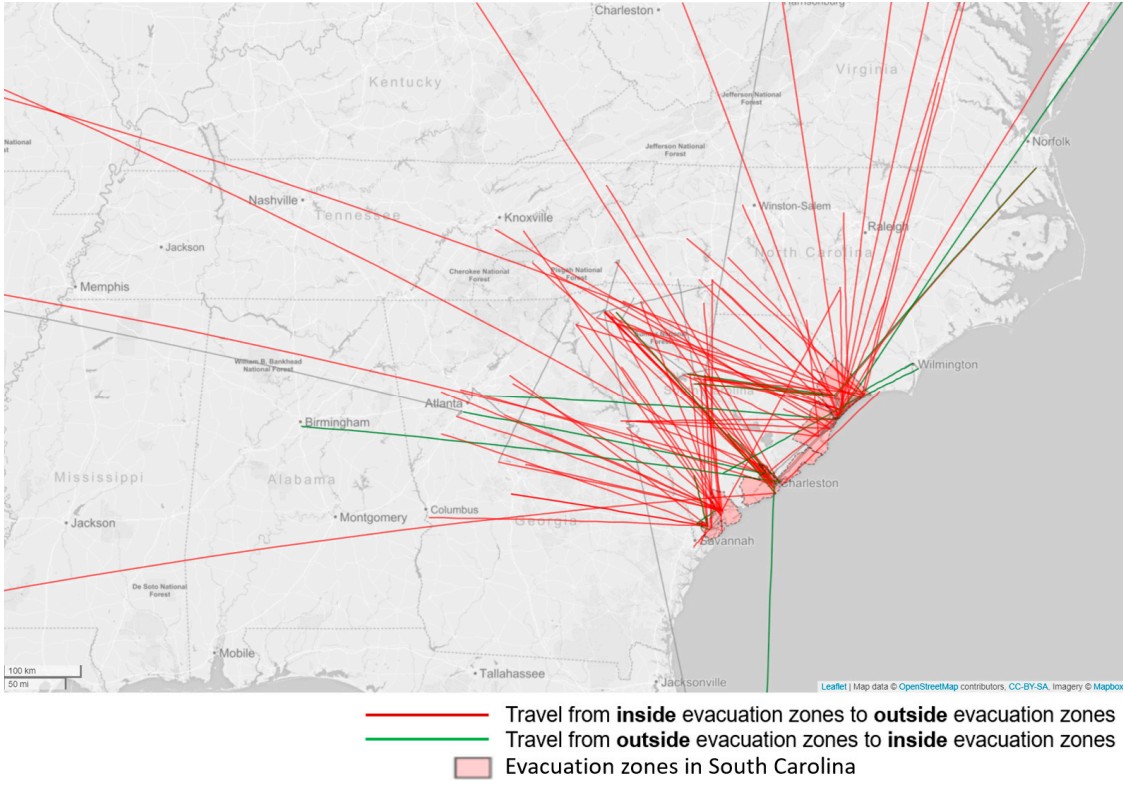

<span style="color:red">———</span> Travel from **inside** evacuation zones to **outside** evacuation zones
<span style="color:green">———</span> Travel from **outside** evacuation zones to **inside** evacuation zones
▢ Evacuation zones in South Carolina

**Figure 8.** Travel direction of evacuation zone residents of South Carolina from October 3, 2016 to October 8, 2016. See Figure 3 to see what the different colored lines represent. The interactive map is available at [36].

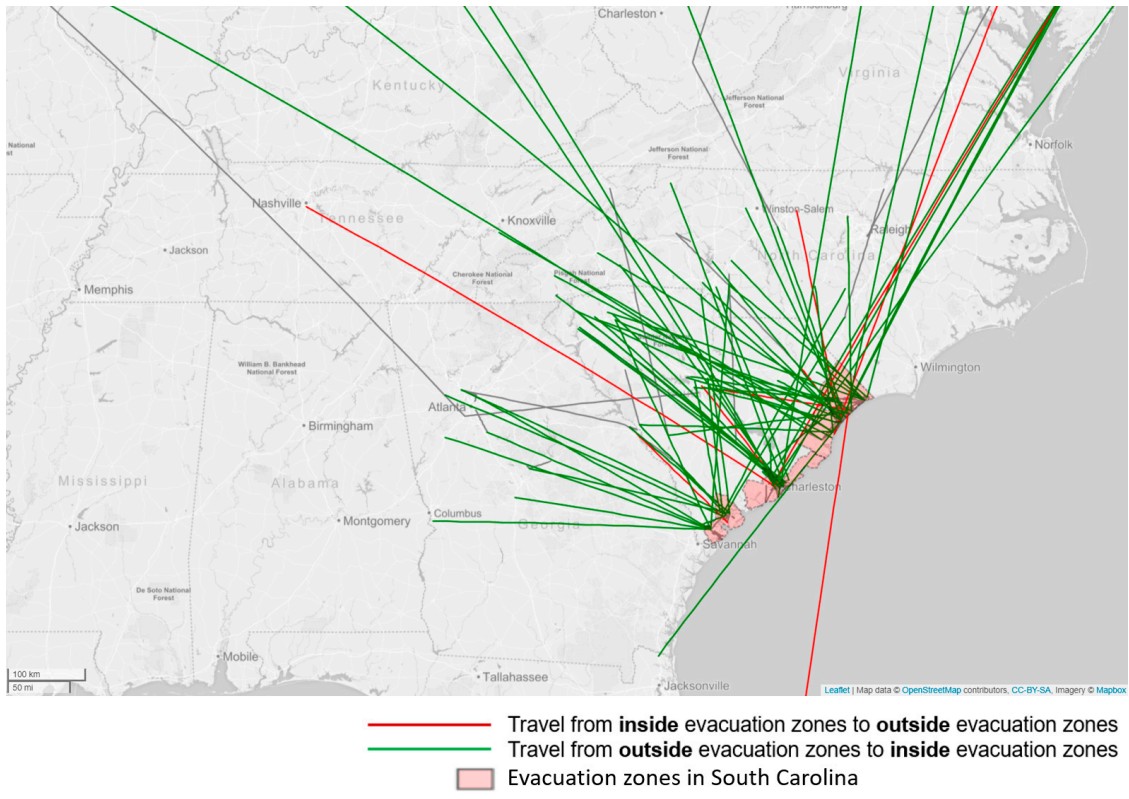

Travel from **inside** evacuation zones to **outside** evacuation zones
Travel from **outside** evacuation zones to **inside** evacuation zones
Evacuation zones in South Carolina

**Figure 9.** Travel direction of evacuation zone residents of South Carolina from October 8 to October 13, 2016. See Figure 3 to see what the different colored lines represent. The interactive map is available at [37].

We also found that EZR in Georgia took a few more days to return from their evacuation trips than those in South Carolina. Similar to the case of SC, in Georgia, most of the travels (about 84%) crossing the boundary of the evacuation zones during the pre-hurricane period were to escape from the evacuation zones (Table 5). Figure 10 visually shows that most Twitter users were escaping from the evacuation zones before or during Hurricane Matthew after the mandatory evacuation order was given to areas east of Interstate 95 (I-95) in six coastal counties (the pink-filled regions) on October 5, 2016. The number of trips out of the evacuation zones during the pre-hurricane period was 127 (Table 5). However, during the post-hurricane period (October 8–13, 2016), the number of returning trips was only 77 (Table 5). Figure 11 visually shows that the number of returning trips (inflow) is much lower than the number of trips leaving the evacuation zones (outflow) in Figure 10. In other words, only 60% of EZR returned to their residential areas from October 8 to 13, 2016 although the evacuation order was lifted on October 9, 2016. However, from October 8 to 16, the number of returning trips (inflow) was 119, which is similar to the number of outflows during the pre-hurricane period (Table 5). Figure 12 visually shows the directions of the returning travel to the evacuation zone for 8 days after the mandatory evacuation order was lifted at 17:00 on Sunday, October 9, 2016.

**Table 5.** The number and the percentage of inflows and outflows in terms of travels of evacuation zone residents in Georgia. See Figure 3 for the definition of inflow, outflow, and other movements.

|  | 10/03/2016–10/08/2016 | 10/08/2016–10/13/2016 | 10/08/2016–10/16/2016 |
|---|---|---|---|
| outflow (leaving) | 127 | 15 | 20 |
| inflow (returning) | 24 | 77 | 119 |
| % outflow | 84.11 | 16.30 | 14.39 |
| % inflow | 15.89 | 83.70 | 85.61 |

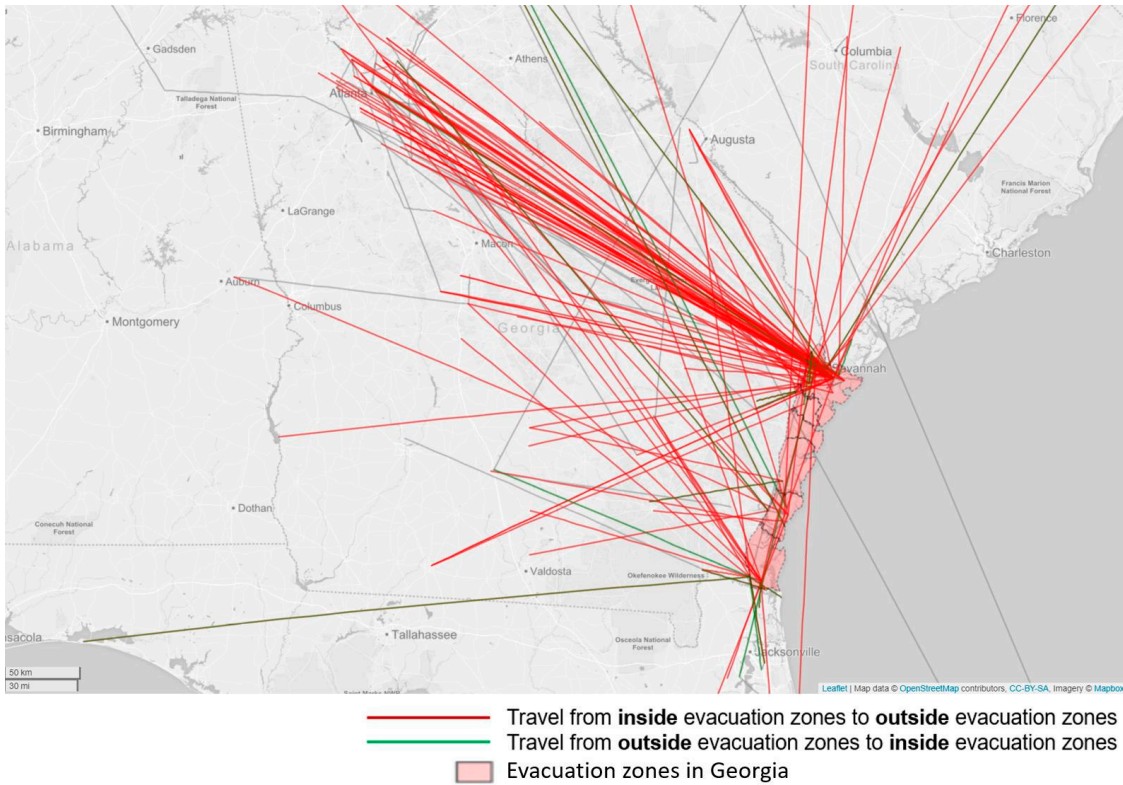

**Figure 10.** Travel direction of evacuation zone residents from October 3 to October 8, 2016. See Figure 3 to see what the different colored lines represent. The interactive map is available at [38].

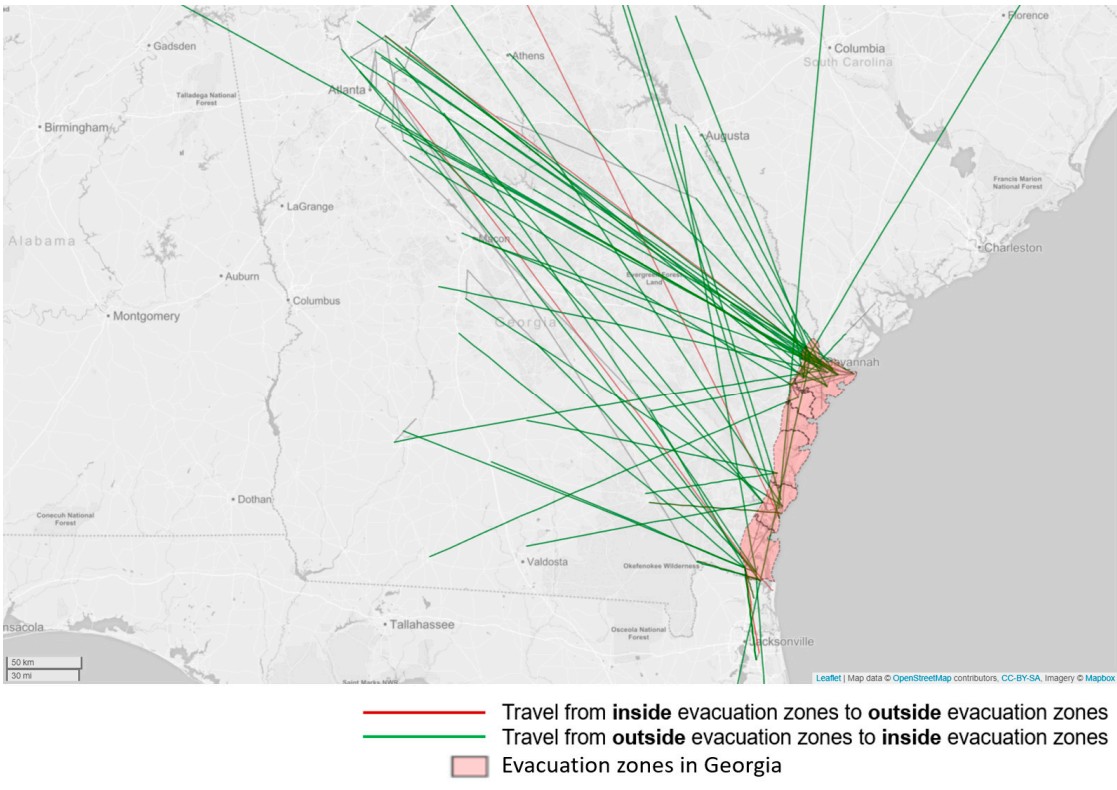

**Figure 11.** Travel direction of evacuation zone residents from October 8 to October 13, 2016. See Figure 3 to see what the different colored of lines represent. The interactive map is available at [39].

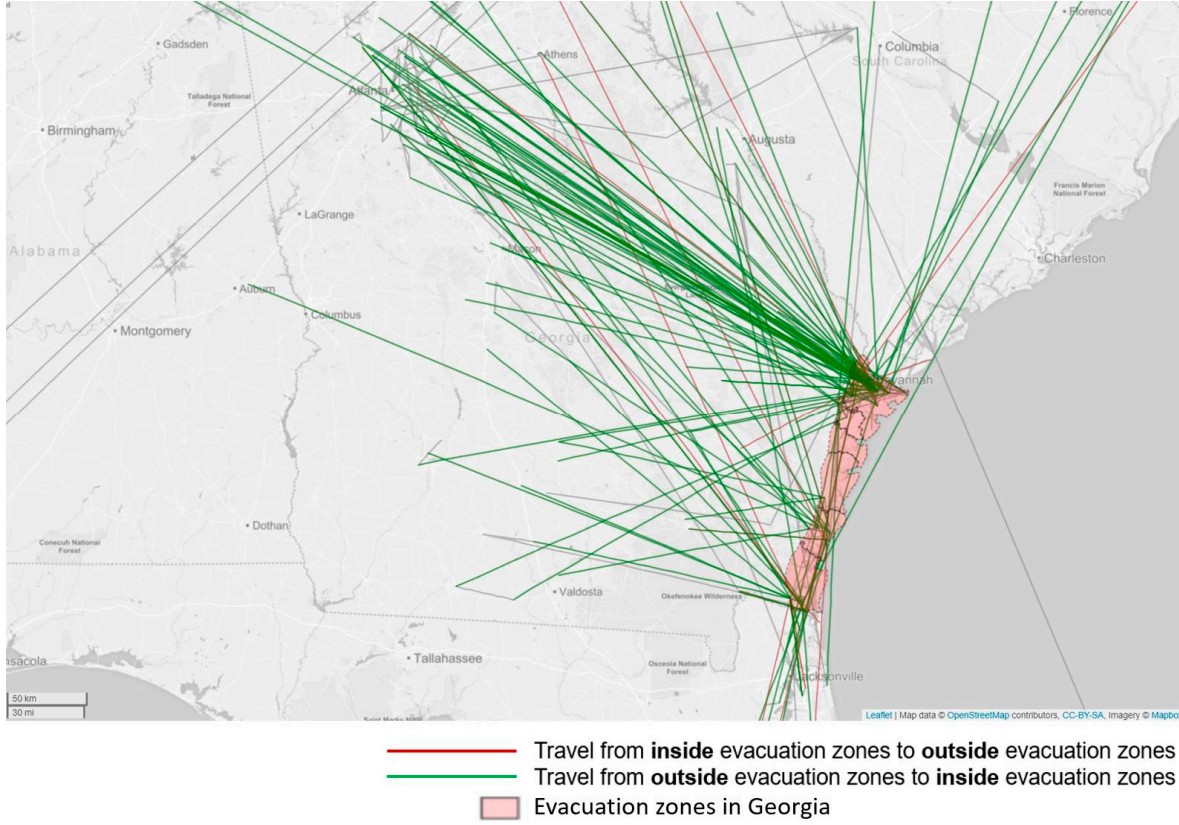

Travel from **inside** evacuation zones to **outside** evacuation zones
Travel from **outside** evacuation zones to **inside** evacuation zones
Evacuation zones in Georgia

**Figure 12.** Travel direction of evacuation zone residents from October 8 to October 16, 2016. See Figure 3 to see what the different colored lines represent. The interactive map is available at [40].

### 4.5. Modeling Human Movements during the Hurricane Evacuation

We examined the distribution of displacement distances between origins and destinations during evacuation travels in both South Carolina and Georgia from October 6, 2016 to October 10, 2016. October 6 was the first day showing more tweets created outside of the evacuation zones than usual in both SC and GA (see Figures 4B and 5B). October 6, 2016 was also the day after mandatory evacuation was ordered in both states. October 10, 2016 was the day after the evacuation order was lifted in both states. At this stage, we only examined movements of Twitter users who traveled outside of the evacuation zones from October 6 to October 10. In other words, we only examined the tweets created by evacuation zone residents who tweeted at least once outside of the evacuation zones during the evacuation period. Based on these tweets, we identified origins and destinations (see Section 3.5.3 to see how they were identified) and then measured the distance of each pair of tweets—i.e., one tweet created inside the evacuation zones (i.e., origin) and the other tweet created outside of the evacuation zones (i.e., destination). Then, we modeled the distribution of displacement distances between each pair of origins and destinations.

Many scholars have noted that travel distances of human movements follow a power-law distribution [6,14–16,41–45]. We intended to examine if human movements during a hurricane evacuation follow the power-law, as usual. Therefore, first, we used the power law (Equation (1)) to model evacuation travels. However, we found that evacuation travel does not follow the power law (Figure S4 in Supplementary Materials shows the result of the power law model fitting). Instead, we found that the displacements are best described by a log-normal distribution (Equation (2)) with the parameters described in Figure 13 [46].

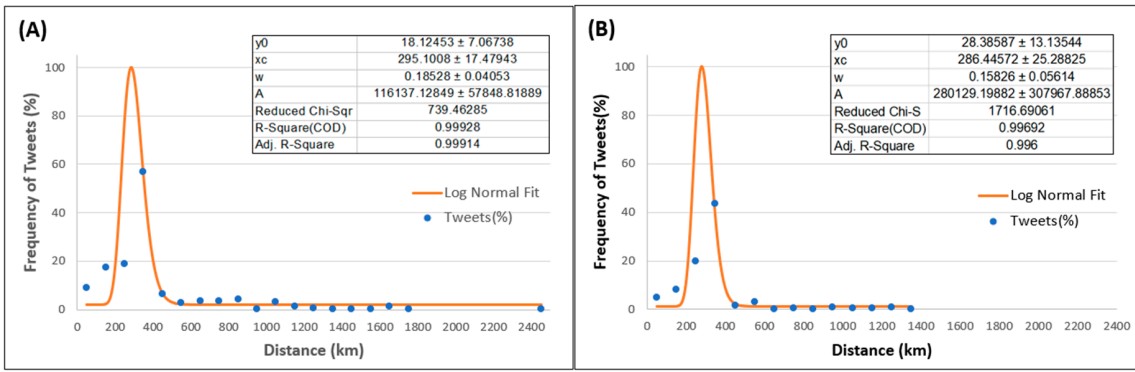

**Figure 13.** Distribution of displacements between origins and destinations during the hurricane evacuation in South Carolina (**A**) and Georgia (**B**) from October 6, 2016 to October 10, 2016 (Tables S2 and S3). Orange line: log-normal fit with characteristic parameters: y0 = offset, xc = center, w = log standard deviation, A = area in Equation (2). R-squared is very close to 1 in both (**A**) and (**B**), which means that the distributions are well approximated by log-normal distribution (Equation (2)).

The log-normal fit curve in Figure 13A,B shows that evacuation zone residents in both states are likely to move more than 200 km but less than 400 km. Figure 13A,B also shows that the highest tweeting frequency is around 290 km (see xc in Figure 13A,B). This finding implies that EZR were most likely to travel about 290 km (3 or 4 h away by car) during the evacuation in both cases of South Carolina and Georgia.

We also found that EZR were likely to visit big cities in the range between 200 and 400 km during the evacuation period. Indeed, 88% of tweets created by EZR in both states were created in metropolitan areas (Figure S5 in Supplementary Materials shows the boundary of each metropolitan area). We counted the number of Twitter users in each metropolitan area and found that the number of Twitter users was likely to be concentrated in highly populated metropolitan areas. The areas where Twitter users were concentrated can be interpreted as frequently visited areas by EZR. Specifically, the top five most visited metropolitan areas by EZR in SC were (1) Greenville including Anderson and Mauldin, SC; (2) Columbia, SC; (3) Charlotte, including Concord-Gastonia, NC–SC; (4) Atlanta, including Sandy Springs and Roswell, GA; and (5) North Charleston, SC (Figure 14). Moreover, 12%, 11%, 10%, 8%, and 8% of EZR in SC visited each of the five metropolitan areas. Greenville, Columbia, Charlotte, and Atlanta are each approximately 320 km, 200 km, 250 km, and 400 km away, respectively, from the evacuation zones in South Carolina. The fifth top visited place was North Charleston, which is a few miles away from Charleston, which was in the evacuation zone. Even though North Charleston is relatively closely located to the evacuation zones in SC compared with Atlanta, EZR of SC traveled to Atlanta as frequently as North Charleston during the evacuation period. Additionally, Greenville, Columbia, and North Charleston are the top 3 most populated cities in South Carolina, except the cities in the evacuation zones. Atlanta in GA and Charlotte in NC are the most populated cities in each of the neighboring states of SC.

On the other hand, EZR of GA overwhelmingly visited Atlanta. During the evacuation period, 44% of EZR in GA visited Atlanta, which is around 350 km away from the evacuation zone. The second top most visited city by EZR of GA during the evacuation was Augusta, which is 250 km away from the evacuation zones. Moreover, 8% of EZR of GA visited Augusta, which is the second most populated city in GA. In other words, more than 50% EZR of GA visited Atlanta or Augusta during the evacuation period.

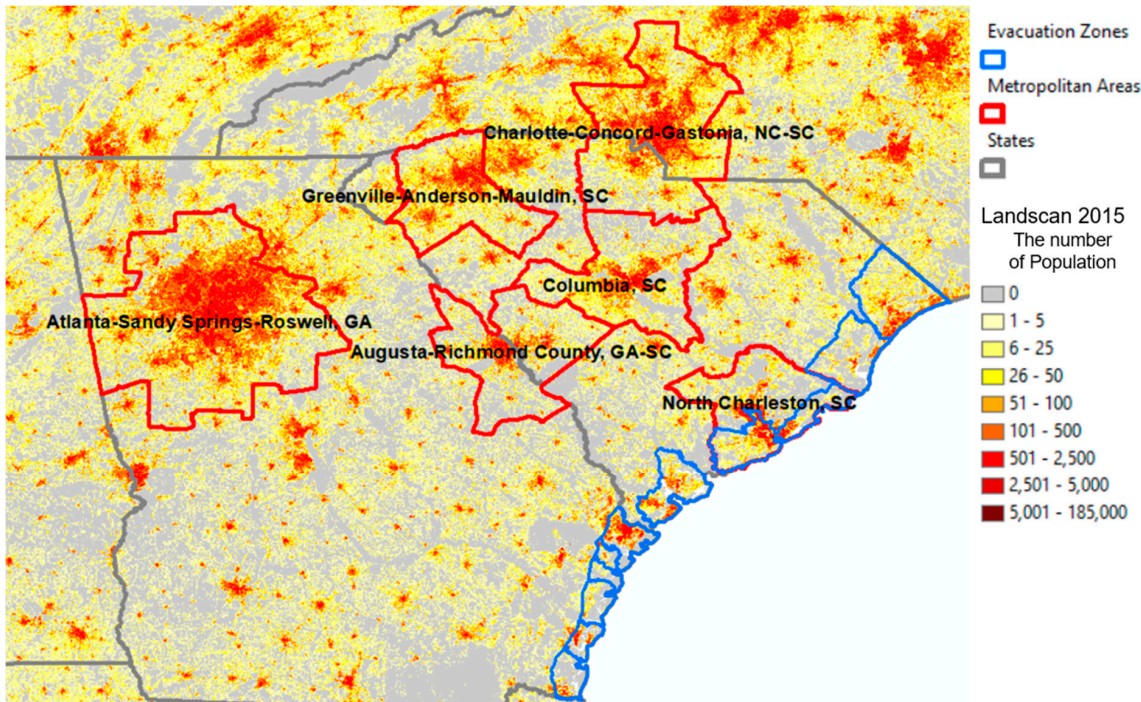

**Figure 14.** Frequently visited metropolitan areas by evacuation zones residents in South Carolina and Georgia.

## 5. Conclusion and Limitation

This study has explored the spatiotemporal aspects of evacuation travel patterns during Hurricane Matthew by using Twitter messages and novel spatiotemporal analytics. These data and analyses have yielded a number of new insights on human behavior during evacuation events. First, the SLIM method for noise filtering helped differentiate between evacuation movements signaled by Twitter messages created by human users among a huge amount of Twitter messages posted by both humans and robots. Second, evacuees among active Twitter users in both South Carolina and Georgia mostly left their town within a few days after they were ordered to evacuate. However, evacuees in these two states showed a different pattern when they made the return trip. After the evacuation order was lifted, evacuees in South Carolina returned home within a few days. However, it took about a week for evacuees in Georgia to return home. Third, evacuees were more likely to choose big cities that are 3 or 4 h away as their destinations for safety rather than nearby small cities during the evacuation. Fourth, the results of model fitting indicate that human movements during the evacuation follow a log-normal distribution instead of the power law distribution, which is seen in general human movement patterns. The visualizations (Figures 6–12) in this study help researchers not only to understand the overall patterns of evacuees' movements (including travel directions, distance, and destination), but also provide some guidance for selecting appropriate models to describe human movements during an evacuation.

We recognize that the trajectories of evacuees identified from Twitter data do not show the complete movement paths of Twitter users. The locations of Twitter users can be traced only when they tweet. If users did not tweet during the evacuation travel, the users' locations could not be known. Furthermore, only small portions of the entire population are Twitter users. In addition, as we identified in Section 4.3, a small portion of Twitter users generate the majority of tweets, while many users tweeted only once during the evacuation period. Thus, the sample population in this study is the active Twitter users who tweet frequently. The sample population considered in this study does not include people who had power outages or disruption to internet services. Therefore, the travel patterns that were identified in this study might not be perfectly representative of all Twitter users who

traveled during the evacuation. More specifically, the estimations in terms of departing and returning time, visited locations during the evacuation travel, and evacuation rate may not be representative.

Due to the limitations of Twitter data mentioned above, researchers should be cautious in generalizing the results of this study to other scenarios. Regarding estimation of evacuation rates, our results are remarkably similar to Cutter et al.'s estimation, (37% versus 35% in 2011), which is considerably lower than the 56% reported by Dow and Cutter in 2002 [47]. In terms of evacuation distance, we found that evacuees are likely to travel 200–400 km, but this distance might not be enough to find safe destinations for those living in the Florida Keys, for example—a string of islands stretching about 200 km off the hurricane-prone southern tip of Florida [48]. In addition, according to Wu et al. [49], the average travel distance during the evacuation was more than 400 km for Hurricane Katrina.

Despite the limitations, this study reveals that the sampled Twitter data can provide a plethora of novel and useful data to explore various evacuees' travel behaviors, such as travel durations, departure–return date, origins, and destinations. These evacuation-related travel behaviors have not been well studied due to the scarcity of data. Previously, the information related to evacuee's travel behaviors was collected through surveys by phone or mails [47–49]. Collecting survey data is time-consuming, expensive, and the number of survey participants is limited. Twitter data, by contrast, provide a much larger sample size, are freely available, and can be collected passively through simple scripts written in Python or R. This study utilized only freely available Twitter data, which is approximately a 1% sample of the entire Twitter data. While this may seem to be a small sample, 1% of the roughly 500 million tweets generated each day is still a considerably larger sample than could be accomplished through traditional survey designs. Furthermore, survey data collection is typically completed a few weeks to a few months after the onset of an emergency, once the evacuation travel has finished. As a result, survey data were not available at the time of the disaster, although real-time changes in evacuee's movements are critical information for disaster relief. Twitter data allow us to observe evacuees' movements in real time during the evacuation period. Thus, a comprehensive lens on evacuees' travel behavior can be found by analyzing social media footprints. Twitter data might not be able to replace survey data, because Twitter data cannot answer qualitative questions (such as "why they did or could not evacuate?"). Although Twitter data cannot provide answers to some questions that surveys can, Twitter data can provide very useful information for real-time disaster management. If so, Twitter data (i.e., passive data) will be able to complement survey data (i.e., active data) during natural disasters.

**Supplementary Materials:** The following are available online at http://sarasen.asuscomm.com/Matthew/Sup, Figure S1: A comparison of tweeting frequency created by evacuation zone residents (EZR) of Florida (FL) before and after noise filtering inside and outside of evacuation zones of FL in the U.S.; Figure S2: A comparison of tweeting frequency created by evacuation zone residents of Florida before and after noise filtering inside and outside of evacuation zones of FL in the U.S.; Figure S3: Distribution of Twitter messages created by each unique author; Figure S4: Distribution of displacements between origins and destinations during the hurricane evacuation in South Carolina and Georgia from October 6, 2016 to October 10, 2016; Figure S5: Metropolitan areas and evacuation zones in South Carolina and Georgia; Table S1: A list of sources creating non-meaningful tweets; Table S2: The number of tweets created by evacuation zone residents of South Carolina for each range of distance; and Table S3: The number of tweets created by evacuation zone residents of Georgia for each range of distance.

**Author Contributions:** Conceptualization, S.Y.H. and M.H.T.; data curation, S.Y.H.; methodology, S.Y.H., G.C., and M.H.T.; software, S.Y.H.; validation, S.Y.H.; formal analysis, S.Y.H.; investigation, S.Y.H.; original draft preparation S.Y.H.; review and editing of manuscript, S.Y.H., M.H.T., E.K., S.R., and G.C.; visualization, S.Y.H.; supervision, S.Y.H.; project administration, S.Y.H.; funding acquisition, M.H.T.

**Funding:** This material is based upon work supported by the U.S. National Science Foundation under IBSS project [grant number 1416509] entitled 'Spatiotemporal Modeling of Human Dynamics Across Social Media and Social Networks' and IMEE project [grant number 1634641] entitled 'Integrated Stage-Based Evacuation with Social Perception Analysis and Dynamic Population Estimation'. Any opinions, findings, and conclusions or recommendations expressed in this material are those of the authors and do not necessarily reflect the views of the National Science Foundation.

**Conflicts of Interest:** The authors declare no conflicts of interest. The funding sponsors had no role in the design of the study; in the collection, analyses, or interpretation of data; in the writing of the manuscript; or in the decision to publish the results.

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
