# Peer review of "How Do Cities Flow in an Emergency? Tracing Human Mobility Patterns during a Natural Disaster with Big Data and Geospatial Data Science"

_urbansci, doi:10.3390/urbansci3020051_

Reviewer 1 Report

This paper aims to identify Twitter users’ evacuation behavior, which is an important topic in emergency management. The paper is written with a clear structure and the topic fits well with the scope of the journal. While I see the merits of this study, I do have some major concerns about the manuscript in its current form and am not quite convinced based on findings provided.

 1. Authors stated in the abstract their two major findings. First is to identify trajectories. In their Figure 3, they showed an example of trajectory of one user. However, in other in-flow and out-flow cases, the authors were not analyzing trajectories but origin-destination pairs. Simply identifying some Twitter users’ location before hurricane, during hurricane, and after hurricane is not considered new scientific findings.

 2. The second findings is drawn from Hurricane Matthew evacuation case study. Whether or not this can be generalized to all evacuation cases is highly questionable. Findings in this case showed in page 20 is very case-specific. In other Hurricane evacuation cases for the similar area, Dow and Cutter (2002) showed 56% evacuees evacuated outside of state. For other areas, such as Florida, 200 – 400km evacuation distance is not enough for those living in Florida Keys (Dash and Morrow, 2000). For Katrina, the average evacuation distance is more than 400km (Wu et al. 2012). Evacuation destination choice is affected by many factors. Without other physical geographical boundaries and social connections, simply drawing conclusion from distance is not convincing.

 3. Authors stated evacuation travel does not follow power law but didn’t provide evidence or statistic test to support this statement. Instead, authors used log-normal distribution. Why was this distribution selected? Any reference to support this choice? This model can be overfitted (noticeably high R-squared).

 4. In the conclusion part, authors summarized 5 major findings from this study. However I don’t think the third one is a scientific finding. It is normal reaction for people living there.

 5. Not clear about the purpose of section 3.2 locating tweets. If you already have tweet coordinates, why need to run bounding box? Besides, the majority of “research design and method” section is about data collection and noise filtering, which, unquestionably, is important, but too distractive and is sort of unbalanced.

 6. Figure 8 – 12, links provided for interactive maps don’t work. These maps are confusing without legend, clear place name label and scale.

References:

Dow, K., & Cutter, S. L. (2002). Emerging hurricane evacuation issues: hurricane Floyd and South Carolina. Natural hazards review3(1), 12-18.

Dash, N., & Morrow, B. H. (2000). Return delays and evacuation order compliance: The case of Hurricane Georges and the Florida Keys. Global Environmental Change Part B: Environmental Hazards2(3), 119-128.

 Wu, H. C., Lindell, M. K., & Prater, C. S. (2012). Logistics of hurricane evacuation in Hurricanes Katrina and Rita. Transportation research part F: traffic psychology and behaviour15(4), 445-461.

 Murray-Tuite, P., Lindell, M. K., Wolshon, B., & Baker, E. J. (2018). Large-Scale Evacuation: The Analysis, Modeling, and Management of Emergency Relocation from Hazardous Areas. CRC Press.

Author Response

The revisions in the manuscript are blue texts. We also proofread and edit grammar errors throughout the whole paper, but these changes could not be traced because all authors work together on google docs. We also revised the Abstract.

Reviewer 2 Report

Thank you for your submission, this is an interesting article that looks at using geotagged Twitter data to examine evacuations due to a hurricane. The interactive maps of the directions is a nice addition, and helps interpret the results.

My overall comment is that I think you need to provide more information about how the users you selected were using Twitter. In table 3 you present the number of Tweets and the number of users, but with Twitter you can have one user that that accounts for 25% of those messages. I may be misinterpreting how you created the density Figures (figure 6 for example). If you are using the total number of tweets, this can be obscured by heavy twitter users. A heavy user can send out 30 tweets from the location outside of the zone and make it look like a hotspot…but it is just one person’s movement to that destination.

With that in mind, and your discussion at the end, I don't think the analysis can justify this advice: Based on the findings in this study, we suggest that policymakers distribute resources accommodating evacuees to big cities that are approximately 200 km and 400 km away from...You don't have a representative sample of the population of evacuees. For example, looking at the direction map you have many users starting in a mid to large city (Myrtle Beach, Savannah, Charleston), just by the nature of who is going to be using Twitter (more likely in larger cities). You may also have missed people who stayed in the evacuation zone, but couldn't tweet because of power outages and disruption to services.

Just some general editorial comments:

Table 2 is split in two pages…You could shorten this table too and only provide one example per source.

Page 13 line 449 : “no other estimation such as a survey result is not available in order” probably should say “no other estimation, such as a survey result, is available in order…”

Reduce the number of classes in the maps (tweet density and population), it’s too many to be useful.

Include the path of Hurricane Matthew at the different time points in the density map.

Author Response

(The authors gave the same response as above.)

Reviewer 3 Report

The paper uses Twitter data to assess the movement of people from disaster-affected areas and back after the disaster period. Despite some minor grammatical errors, I found it to be an interesting study and use of Twitter data. I have a few suggestions for improving the paper;

1) I found some table and figure captions to be too long and thought some descriptions could be moved to the text e.g. Table 3.

2) Table 3 has two rows labeled (d).

3) Table 3: I suggest starting with meaningful tweets on first row for a more logical flow.

4) Line 325: Is it necessary to repeat the methodological process here?

5) Line 626: the authors hint that the results of their study may not be representative of the entire population. I have a question from this claim: what is the value of a study that is not representative and can not be generalised to the rest of the population?

6) I think the authors need to revise the justification and value of Tweeter data. Line 629-631: The travel behaviours (travel duration, dates, origins, destinations) from tweets listed by the authors are questionable because as they (authors) have acknowledged elsewhere in the paper, Tweeter data depends on the people actually tweeting. So, a date, for example, will merely show when the tweet, and not the travel, was made. Similarly, the location will be of the tweet and may not necessarily be an origin or final destination. Social media provide very useful data, and the authors seem to be understating and misstating its value here.

Author Response

The revisions in the manuscript are blue texts. We also proofread and made edits on grammar errors throughout the whole paper, but these changes could not be traced because all authors work together on google docs. We also revised the Abstract.

Round  2

Reviewer 1 Report

The authors have addressed my concerns in this revision. I would recommend for publication.